



# Dehydration and low ozone in the tropopause layer over the Asian monsoon caused by tropical cyclones: Lagrangian transport calculations using ERA-Interim and ERA5 reanalysis data

Dan Li[1,2], Bärbel Vogel[1], Rolf Müller[1], Jianchun Bian[2,3], Gebhard Günther[1], Felix Plöger[1], Qian Li[2], Jinqiang Zhang[2,3], Zhixuan Bai[2], Holger Vömel[4], and Martin Riese[1]

[1]Institute of Energy and Climate Research: Stratosphere (IEK-7), Forschungszentrum Jülich, Jülich, Germany
[2]Key Laboratory of Middle Atmosphere and Global Environment Observation (LAGEO), Institute of Atmospheric Physics, Chinese Academy of Sciences, Beijing, China
[3]College of Earth Science, University of Chinese Academy of Sciences, Beijing, China
[4]Earth Observing Laboratory, National Center for Atmospheric Research, Boulder, CO, USA

**Correspondence:** Dan Li (lidan@mail.iap.ac.cn) and Jianchun Bian (bjc@mail.iap.ac.cn)

**Abstract.** Low ozone and low water vapour values near the tropopause over Kunming, China were observed using balloon-borne measurements performed during the SWOP (sounding water vapour, ozone, and particle) campaign in August 2009 and 2015. Here, we investigate low ozone and water vapour signatures in the upper troposphere and lower stratosphere (UTLS) using FengYun-2D, FengYun-2G, Aura Microwave Limb Sounder (MLS) satellite measurements and backward trajectory

calculations driven by both ERA-Interim and ERA5 reanalysis data. Trajectories with kinematic and diabatic vertical velocities were calculated using the Chemical Lagrangian Model of the Stratosphere (CLaMS) trajectory module.

All trajectory calculations show that air parcels with low ozone and low water vapour values in the UTLS over Kunming measured by balloon-borne instruments originate from the western Pacific boundary layer. Deep convection associated with tropical cyclones over the western Pacific transports boundary air parcels with low ozone into the cold tropopause region.

Subsequently, these air parcels are mixed into the strong easterlies on the southern side of the Asian summer monsoon anticyclone. Air parcels are dehydrated when passing the lowest temperature region ($<190\,\mathrm{K}$) over the western Pacific during quasi-horizontal advection. However, trajectory calculations show different vertical transport via deep convection depending on the employed reanalysis data (ERA-Interim, ERA5) and vertical velocities (diabatic, kinematic). Both the kinematic and the diabatic trajectory calculations using ERA5 data show faster and stronger vertical transport than ERA-Interim primarily due to

ERA5's better spatial and temporal resolution, likely resolving more convective events.

## 1 Introduction

The Asian summer monsoon (ASM) anticyclone plays an important role in transporting air masses from the troposphere into the stratosphere (e.g. Chen et al., 2012; Garny and Randel, 2016; Randel et al., 2010; Vogel et al., 2019). Air masses within the ASM anticyclone are impacted by surface sources from the western Pacific, India, Southeast Asia and the Tibetan

Plateau (Bergman et al., 2013; Park et al., 2008; Tissier and Legras, 2016; Vogel et al., 2015). Recently, Höpfner et al. (2019)





have reported observations that indicate upward transport from ground sources of ammonia ($NH_3$) to greater altitudes with subsequent formation of ammonium nitrate ($NH_4NO_3$) during further ascent in the ASM anticyclone. This is further confirmed by satellite observations of air pollution from biomass burning (e.g. carbon monoxide, hydrogen cyanide, ethane) which clearly support the idea of convective uplift from the surface to the tropopause layer within the ASM anticyclone (Park et al., 2008;

Yan and Bian, 2015). Using trajectory calculations, Chen et al. (2012) demonstrate that 38% of air masses at tropopause height within the ASM region are from the western Pacific region and South China Sea. Bergman et al. (2013) also show that 36.1% of air parcels at 100 hPa originate from the western Pacific based on diabatic trajectory calculations.

Convection in the tropical cyclones (e.g. typhoons in the western Pacific) can lift air masses from the marine boundary layer into the tropopause layer (Li et al., 2017; Minschwaner et al., 2015; Pan et al., 2016; Vogel et al., 2014). Tropospheric

ozone can be decomposed by halogen induced photochemical reactions in the tropical marine boundary layer (von Glasow et al., 2002). As a result, low ozone values are distributed over at least several thousand kilometers. Low ozone values are sometimes measured in the upper troposphere within typhoons or hurricanes (Cairo et al., 2008; Fu et al., 2013), due to fast vertical transport caused by tropical cyclones. Minschwaner et al. (2015) find that hurricane Henriette uplifted air with low ozone from the boundary layer in the central and eastern Pacific to the upper troposphere with subsequent transport to Socorro

(North America) according to balloon-borne ozonesonde and satellite measurements. Newton et al. (2018), using aircraft measurements in Guam, show that low ozone values (20 ppbv) in the tropical tropopause layer (TTL) are caused by deep convection, which has lifted ozone-poor boundary-layer air to the TTL over the western Pacific in 2014.

The lowest temperatures in the TTL are often found over the western Pacific. When air parcels are transported horizontally via the lowest temperature tropopause region (cold trap) over the western Pacific in winter, dehydration occurs (Holton and

Gettelman, 2001). Read et al. (2008), using model calculations and the Aura Microwave Limb Sounder (MLS) measurements, show that water vapour is controlled by the lowest temperature in the TTL, that is called, in-situ freeze-drying.

In-situ observations from the Soundings of Ozone and Water in the Equatorial Region (SOWER) campaign in winter show a clear correspondence between dry air parcels and low temperatures during advection in the TTL over the western Pacific (Hasebe et al., 2007). The match technique was applied to quantify the features of dehydration or hydration for air parcels in

the TTL (Hasebe et al., 2013; Inai et al., 2013). The relative humidity with respect to ice ($RH_i$) of 146±19%, a threshold of ice nucleation, indicates the development of ice clouds. Dehydration is ongoing until $RH_i$ decreases to 75±23% between 350 K to 360 K (Inai et al., 2013). Cold-trap dehydration is accompanied by slow ascent in the TTL and quasi-horizontal advection between 360 K and 380 K (Hasebe et al., 2013). Lagrangian trajectory simulations also show that water vapour mixing ratios transported into the stratosphere from the South China and Philippine Seas are lower compared to South Asian subcontinent

and the Southern Slope of the Himalayas and the Tibetan Plateau (Wright et al., 2011). Deep convection contributes to low temperature regions in the subtropical lower stratosphere, and leads to a dry stratosphere (Randel et al., 2015).

Low water vapour mixing ratios below 2 ppmv were observed at the cold point tropopause (370–380 K) during the Stratospheric-Climate Links with Emphasis on the Upper Troposphere and Lower Stratosphere (SCOUT-O3) air craft campaign in November and December 2005 over Darwin, Australia (Schiller et al., 2009). These low water vapour mixing ratios are primarily deter-





mined by effective freezing-drying near the tropopause caused by deep convection associated with the cumulonimbus system Hector.

Li et al. (2017) found that tropical cyclones that occurred over the western Pacific uplifted marine boundary layer air masses with low ozone to the ASM anticyclone, using balloon measurements and trajectory calculations. In general, typhoons decrease ozone values near the tropopause over the western Pacific. However, hitherto the variability of water vapour concentrations in

the upper troposphere and lower stratosphere (UTLS) region has not been analysed in detail. Using balloon-borne measurements, MLS data, and the Chemical Lagrangian Model of the Stratosphere (CLaMS) trajectory calculations, we will investigate how the tropical cyclones impact water vapour structures in the UTLS region over Kunming. This paper is organized as follows: Sect. 2 describes the balloon measurement data and the trajectory calculations with the CLaMS model. In Sect.3, we will present two-case studies where dehydration is found in the measurements. A comparison of diabatic and kinematic trajectory

calculations driven by both ERA-Interim and ERA5 reanalysis data will be presented as well. A summary will be given in the final section.

## 2    Measurements and trajectory calculations

### 2.1    Balloon-borne measurements

Vertical profiles of temperature, ozone, and water vapour were measured over Kunming (25.01° N, 102.65° E, above sea level

(asl.) 1889 m), China, in August 2009 and 2015 during the SWOP (sounding water vapour, ozone, and particle) campaign (e.g. Bian et al., 2012; Li et al., 2017, 2018). In 2009, 11 balloons were launched during the period 7–13 August; seven and four balloons were launched in the daytime and at night (around 10:00 and 22:30 local time, UTC+8). Detailed information about the launches is provided by Bian et al. (2012, see table S1). In 2015, 12 balloons were launched around 22:30 (local time, UTC+8) from 3 to 18 August. Profiles of temperature, pressure, relative humidity, and winds were measured by Vaisala RS80

radiosondes in 2009 and iMet radiosondes in 2015. Profiles of ozone and water vapour mixing ratios were measured by an electrochemical concentration cell (ECC) ozonesonde (Komhyr et al., 1995) and a cryogenic frost point hygrometer (CFH) (Vömel et al., 2007), respectively. Data are transmitted to Kunming ground receiving station ∼100 minutes and are recorded each second. The ozone and water vapour precision in the troposphere is 10% and 5%, respectively (Vömel et al., 2016).

The (saturation) water vapour mixing ratio is calculated from the (ambient) frost point temperature using the Hyland–Wexler

equation (Hyland and Wexler, 1983) for liquid water and from the Goff–Gratch equation (Goff and Gratch, 1946; Murphy and Koop, 2005) for ice water. The match technique, means that the same air parcel is observed more than once (Hasebe et al., 2013; Inai et al., 2013). Then the difference of water vapour values between these observation times was applied to quantify the features of dehydration or hydration for air parcels during horizontal advection in the TTL.

Ozone profiles for Naha (26.21° N, 127.69° E, asl. 28.1 m), Okinawa Island, Japan during the period of 2008–2017 were

obtained from the World Ozone and Ultraviolet Radiation Data Centre (Naja and Akimoto, 2004). Naha station is located approximately 2500 km east of Kunming, and these two sites are nearly on the same zonal line. Profiles on 4 August 2009, 05:30 UTC and on 5 August 2015 from Naha will be presented in sections 3.1.2 and 3.2.2, respectively.





The upper air soundings of Chenzhou, Ganzhou, Xiamen, Taipei, and Ishigakijima in August 2009 and Haikou, Wuzhou, Hongkong, Shantou, Laoag, and Ishigakijima in August 2015 were used in this paper. The data are downloaded from the University of Wyoming. Balloons were launched routinely, every twice per day at 00:00 UTC and 12:00 UTC. They provide profiles of temperature, pressure, relative humidity, and wind vector from the surface to 30 km.

## 2.2 Satellite data

Microwave Limb Sounder (MLS) measurements are used to validate ozone and water vapour profiles in the tropopause layer over the western Pacific. MLS level 2 version 4.2x standard atmospheric product data are used for 4 August 2009 and the period of 31 July–10 August 2015. MLS provides ozone with 10 pressure levels ranging from 261 hPa to 46 hPa (9.5–21.5 km) along the orbit track (Livesey et al., 2018). Precision on individual ozone profiles is within 5% or better in the lower stratosphere, and is approximately 15%–25% at 100 hPa, while with poor behavior at low latitudes in the upper troposphere. The mean deviation is lower than 20% between the MLS and the ECC ozone values in the upper troposphere over the Tibetan Plateau (Shi et al., 2017).

The FengYun-2D ("Feng and Yun" means "winds and clouds" in Chinese), or FY-2D in acronym and FY-2G are the geostationary meteorological satellite series of China, organized and operated by the national satellite meteorological center of CMA (China Meteorological Administration). FY-2D was launched on 8 December 2006, and carried a payload with a five-channel Stretched Visible and Infrared Spin Scan Radiometer (S-VISSR) to track cloud motion. Long wave infrared (10.3–11.3 $\mu$m) with a spatial resolution of 5 km is used to detect cloud moving. Here, we used the FY-2D blackbody brightness temperature (TBB) product to determine the cloud top temperature. FY-2G was launched on 31 December 2014 and carried a S-VISSR. We use cloud top temperature (CTT) to detect the cloud motion and location.

## 2.3 Trajectory calculations based on ERA-Interim and ERA5 reanalysis data

Diabatic and kinematic backward trajectories along two balloon's ascending paths over Kunming on 8 August 2009 and 10 August 2015 were calculated using the CLaMS trajectory module (Konopka et al., 2004; McKenna et al., 2002; Pommrich et al., 2014). The CLaMS kinematic trajectory calculations employ pressure ($p$) as the vertical coordinate and omega ($\omega$) as vertical velocity. In contrast, the diabatic trajectory calculations employ a hybrid $\sigma$-potential temperature coordinate (Pommrich et al., 2014) and the diabatic vertical velocity derived from total heating rates. Note, that above $\sigma = p/p_s$=0.3, hence throughout the stratosphere and upper troposphere, the vertical coordinate is just potential temperature. Diabatic calculations have important advantages for the upper troposphere and stratosphere (e.g. Ploeger et al., 2010, 2011; Schoeberl et al., 2003; Schoeberl and Dessler, 2011). The dynamic fields from the European Centre for Medium-range Weather Forecasts (ECMWF) interim reanalysis (ERA-Interim) (Dee et al., 2011) and ECMWF's next-generation reanalysis ERA5 (Hersbach and Dee, 2016) are used to drive the CLaMS model. ERA-Interim input data is recorded on a $1° \times 1°$ grid every 6 hours, on 60 hybrid levels from about 1013.25 hPa to 0.1 hPa. The ERA5 input data is provided on a $0.3° \times 0.3°$ grid every hour on 137 hybrid levels from the surface to 0.01 hPa. The ERA5 has much higher spatial and temporal resolution than ERA-Interim, especially in the upper troposphere and lower stratosphere (Hoffmann et al., 2019, see Fig. 1). The horizontal wind fields show similar





structure at 150 hPa between ERA-Interim and ERA5 reanalysis data (Fig. A1). The vertical velocities from ERA5 show finer spiral structures compared to that from ERA-Interim at 150 hPa in regions of tropical cyclones (Fig. 1). Stronger negative $\omega$ corresponds to ascending motion around the convection centers associated with tropical cyclones as shown in the right column of Fig. 1. The outflow of deep convection is clearly much stronger on 6 August 2009 than on other days based on ERA5 reanalysis data.

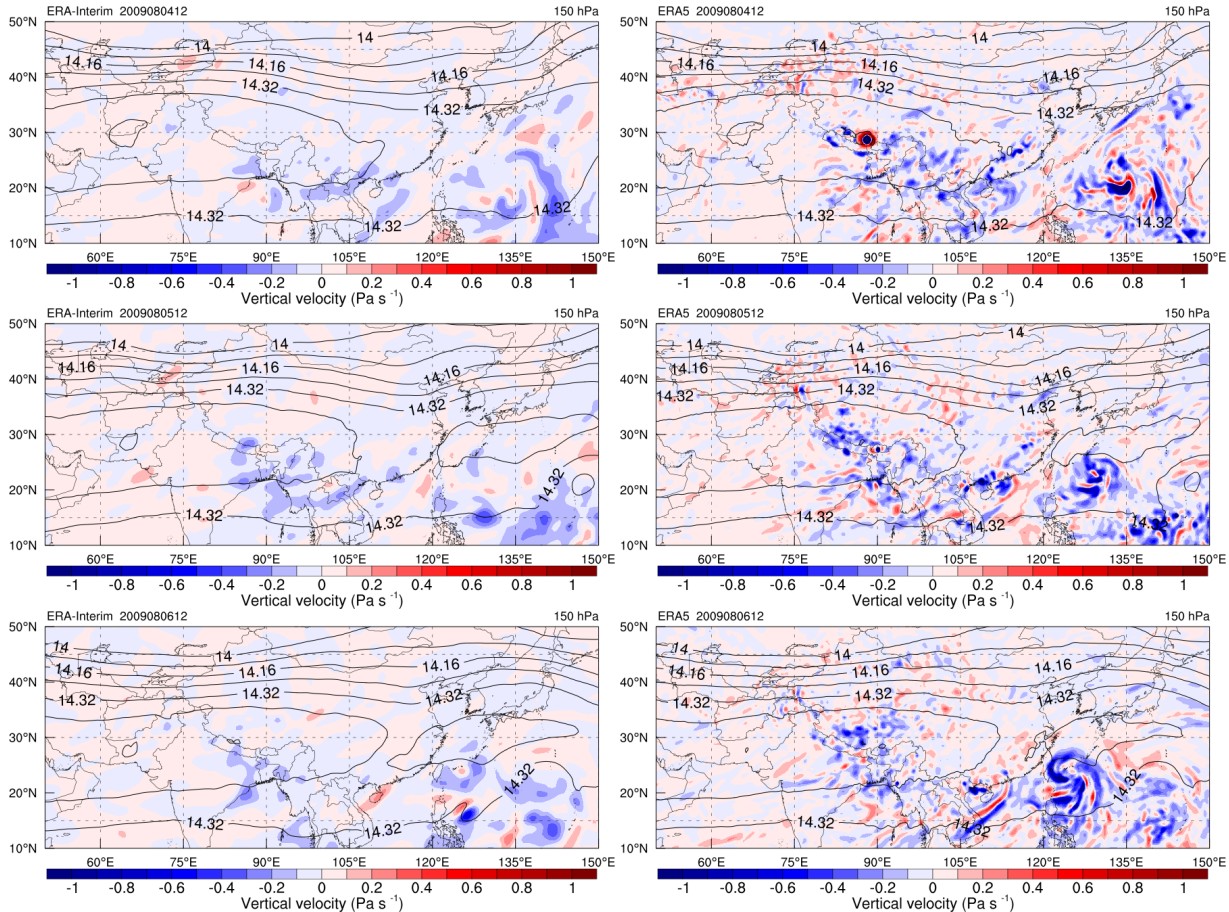

**Figure 1.** Geopotential height (black lines, gpkm) and vertical $\omega$ velocities (shaded, $\mathrm{Pa\,s^{-1}}$) at 150 hPa on 4, 5, and 6 August 2009 from ERA-Interim (left) and ERA5 (right).






## 3  Results

### 3.1  Case 1: 8 August 2009

All vertical profiles of temperature, ozone, and water vapour mixing ratios over Kunming in August 2009 are shown in Fig.
2. The minimum temperature of 192.3 K occurs on 8 August (Fig. 2a). The lapse rate tropopause potential temperature on 8

August is located at 381.3 K. Ozone mixing ratios in the upper troposphere ranged from 40 ppbv to 120 ppbv in August 2009.
Ozone mixing ratios observed on 8 August show low values in the potential temperature layer between 376 K and 384 K, with
a minimum value of 35 ppbv around 378 K (Fig. 2b). Simultaneously, water vapour near the tropopause on 8 August is also
very low, around 2.5 ppmv at 378 K (Fig. 2c). Low ozone and low water vapour mixing ratios near the tropopause on 8 August
are strongly correlated, marked by the shaded regions in Fig. 2. It is necessary to mention that Bian et al. (2012) highlight this

low ozone and low water vapour values using the same observation data. They argue that the rapid convective uplift and the
low temperature near the tropopause are the reason for these coinciding low ozone and low water vapour values. However, they
only offer limited explanation on the detailed reasons, which are investigated and discussed in the following subsections.

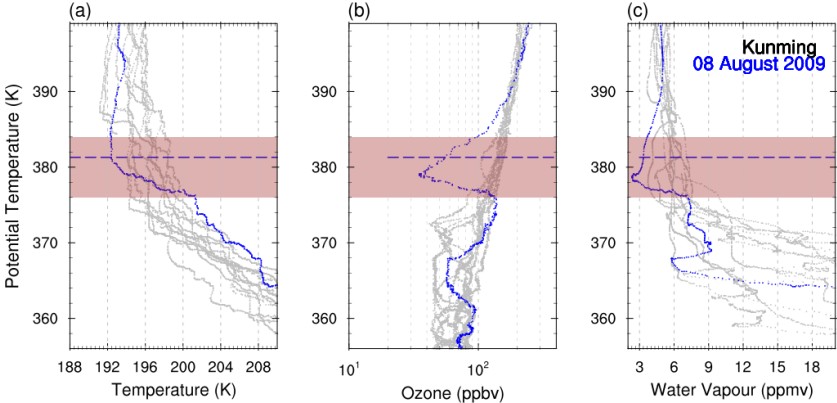

**Figure 2.** All vertical profiles of (a) temperature, (b) ozone mixing ratios (OMRs), and (c) water vapour over Kunming in August 2009. The
profile on 8 August is marked as blue line. The horizontal blue dashed line marks the lapse rate tropopause on 8 August. The shading region
denotes the low ozone and low water vapour mixing ratios region.

### 3.1.1  Background meteorology

Figure 3 shows the 5-day average (4–8 August 2009) of the geopotential height isolines (>16.7 gpkm) at the 100 hPa pressure

level with the temperature (shaded), horizontal wind vector, and tracks of typhoon Morakot from the ERA-Interim reanalysis
data. Isolines are used to denote the scope of the ASM anticyclone. Morakot formed early on 2 August, within a monsoon
trough about 1000 km east of the Philippines. Morakot moved westward since 4 August, and arrived at the south of Naha on
6 August. During the period 4–8 August 2009, the southeastern edge of the ASM anticyclone was located above the top of





the typhoon and covered the lowest temperature region (<192 K). Similar results can also be obtained from the ERA5 analysis data (not shown).

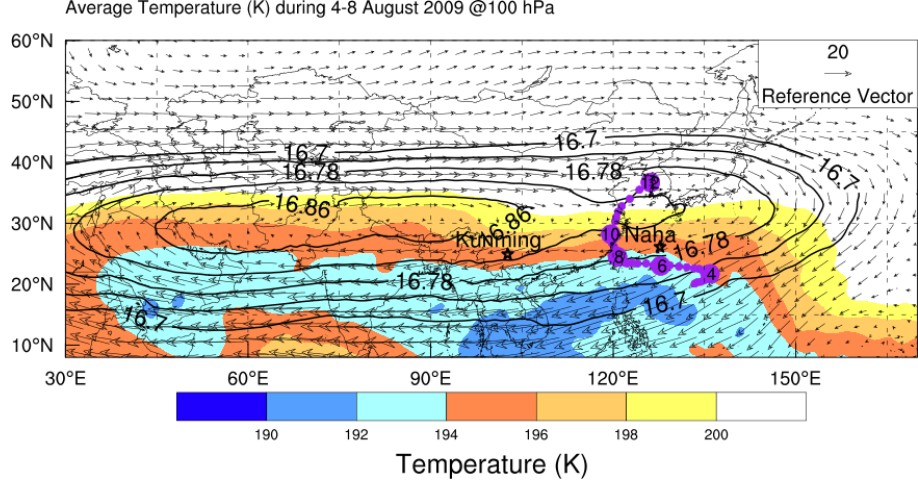

**Figure 3.** (a) The 5-day average (4–8 August) of geopotential height (black lines, >16.70 gpkm), temperature (shaded, K), and wind speed (vectors, m s$^{-1}$) from ERA-Interim at the 100 hPa pressure with the tracks of typhoon Morakot (purple dots). The numbers in purple dots indicate days in August 2009. The black stars mark the locations of Naha and Kunming.


### 3.1.2 Low ozone in the tropopause layer

A balloon was launched in Naha, Japan on 4 August 2009 before the typhoon Morakot passed through this site. Low ozone values (35 ppbv) appeared at the layer between 360 K and 370 K (Fig. 4a). The observed mean ozone value of 10 years (2008–2017) in this layer is about 80–120 ppbv. The balloon on 8 August in Kunming also captured the extremely low ozone structure,
but at the layer between ∼376–384 K. The altitude of the low ozone appearing in Kunming on 8 August is higher than that in Naha on 4 August 2009. Low ozone values near the tropopause within the ASM anticyclone were also found in Lhasa in 2013 (Li et al., 2017), which are caused by the combination of the rapid vertical transport from typhoon convection and the horizontal transport at the edge of the ASM anticyclone. The ERA-Interim and ERA5 temperature and ozone data were interpolated to the balloon tracks on 8 August. Note that the balloon data were not assimilated into the ERA5 and ERA-Interim
reanalysis. Compared to balloon observation, the ERA5 temperature profile shows a lower value than ERA-Interim temperature on 8 August 2009. Both the ERA-Interim and ERA5 ozone profiles show large positive deviations in the tropopause layer with respect to observations.

### 3.1.3 Low temperatures and dehydration in the western Pacific

Figure 5 shows 7-day backward trajectories of air parcels with low ozone and low water vapour between 376 K and 384 K
initialized on 8 August 2009 in Kunming. A comparison of diabatic and kinematic trajectory calculations is shown based on




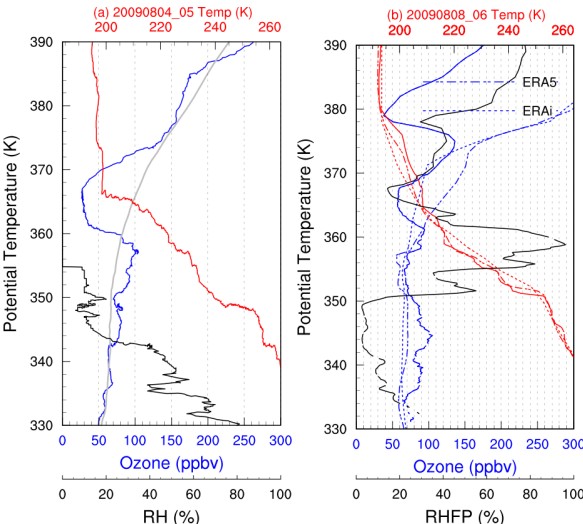

**Figure 4.** Profiles of ozone (blue), mean ozone (grey), relative humidity (RH from radiosondes, and RHFP from CFH, black), and temperature (red) in (a) Naha on 4 August and (b) Kunming on 8 August (solid line–observation, dashed line–ERA-interim, dash-dotted line–ERA5).

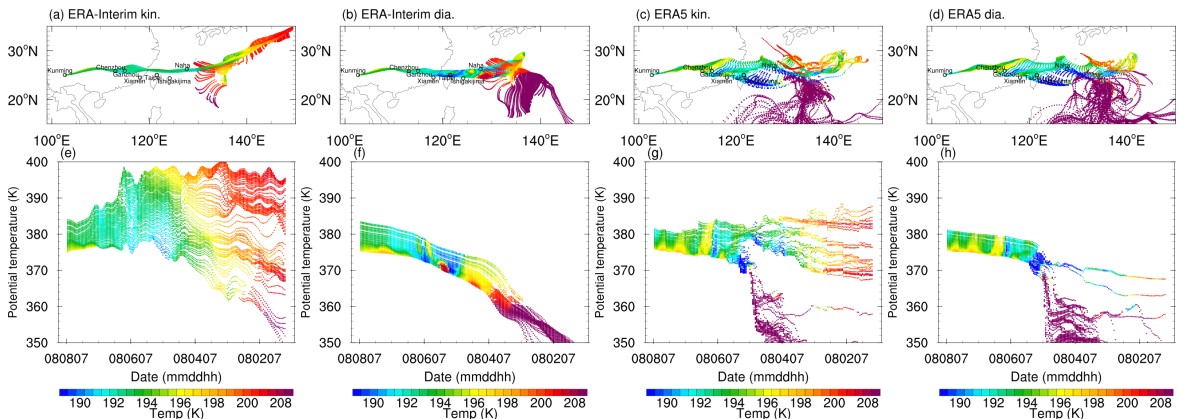

**Figure 5.** ERA-Interim kinematic (a, e) and diabatic (b, f), and ERA5 kinematic (c, g) and diabatic (d, h) backward trajectories for air parcels started along the measured balloon profile between 376 K and 384 K colour-coded by temperature in (top) longitude–latitude cross section and (bottom) as a function of time vs. potential temperature.

ERA-Interim and ERA5 reanalysis data. Air parcels originating from the western Pacific were transported to the tropopause layer under significant influence of typhoon convection, then affected by the easterly wind flow at the south flank of the ASM anticyclone. Subsequently, air parcels moved horizontally via Naha, Ishigakijima, Taipei, Xiamen, Ganzhou, Chenzhou, then to Kunming according to backward trajectories projected on the map (Figs. 5a–d). The convection associated with typhoon Morakot transported ozone-poor air from the marine boundary layer to the tropopause layer, these low ozone values were captured as these air parcels moved westward to Naha and Kunming. Trajectories from ERA-Interim and ERA5 reanalysis data



show the lowest temperature region ($<190\,\mathrm{K}$) in the tropopause layer over Taiwan and Ishigakijima except the ERA-Interim kinematic trajectories. The main difference of the backward trajectories is the vertical transport over the western Pacific, where tropical cyclones may occur (Figs. 5e–h). Large-scale slow descent processes ($\sim$1–2 K/day) can only be seen clearly from

ERA-Interim kinematic trajectories (Fig. 5e). This is consistent with Ploeger et al. (2011) who showed much stronger vertical dispersion for ERA-Interim kinematic trajectories. Although all of the backward trajectories display the vertical transport within deep convection, the timescale and the strength of air parcels' vertical transport processes are different according to the backward trajectories as a function of time and isentrope. The timescale of the vertical transport from the lower troposphere to the tropopause layer based on ERA-Interim diabatic trajectories is about 4 days from 1 August to 5 August (Fig. 5f). In contrast,

the vertical transport timescale based on the ERA5 kinematic and diabatic trajectory calculations is 2 days around 4–5 August (Figs. 5g and h). Both the kinematic and diabatic trajectories from ERA5 display faster vertical transport than ERA-Interim. It is very likely that ERA5 resolves more convective events (e.g. Fig. 1), due to its better spatial and temporal resolution. Hoffmann et al. (2019) have also shown that tropical cyclones are represented better in ERA5, compared to ERA-Interim reanalysis data.

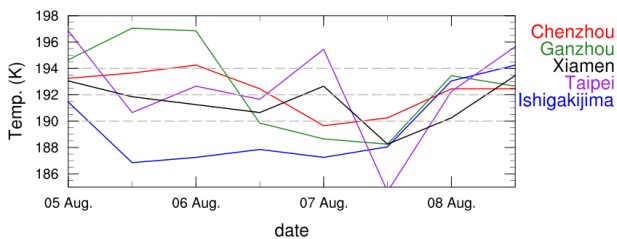

**Figure 6.** The time series of the lowest temperature in the tropopause layer for Chenzhou, Ganzhou, Xiamen, Taipei, and Ishigakijima.

The time series of the lowest temperature from Chenzhou, Ganzhou, Xiamen, Taipei, and Ishigakijima are shown in Fig. 6 based on upper air soundings. The balloon measurement at Taipei captured the lowest temperature (186 K) during the period 5–7 August 2009 near the tropopause. ERA-Interim kinematic trajectories missed the lowest temperature values compared to the upper air soundings, and do not show the strong updraft in tropical cyclones. Therefore, ERA-Interim kinematic trajectories will not be considered further in the following.

Figure 7 shows the blackbody brightness temperature from infrared radiation (IR) channel of FY-2D. The blue colour marks the low brightness temperature region, which means that deep convection with high cloud top occurred in the centre of a tropical cyclone. Air parcels were located right above deep convection of typhoon Morakot on 4 August according to the ERA5 kinematic and diabatic trajectories (Fig. 7a). The ERA-Interim diabatic trajectories were located further at the northern edge of the typhoon. One day later, all trajectories arrived at Naha site over the western Pacific region (Fig. 7b). Air parcels

continue to move westward towards Kunming under clear sky conditions during the last two days (Figs. 7c and d).

In order to investigate the variation of water vapour mixing ratios at the top of the typhoon convection, Figs. 8a and b show the temperature along the 4-day backward trajectories of air parcels at 376–384 K on 8 August 2009 with the brightness temperature from FY-2D (grey dots). The y-axis of Figure 8 represents a vertical altitude coordinate (decreasing temperatures).

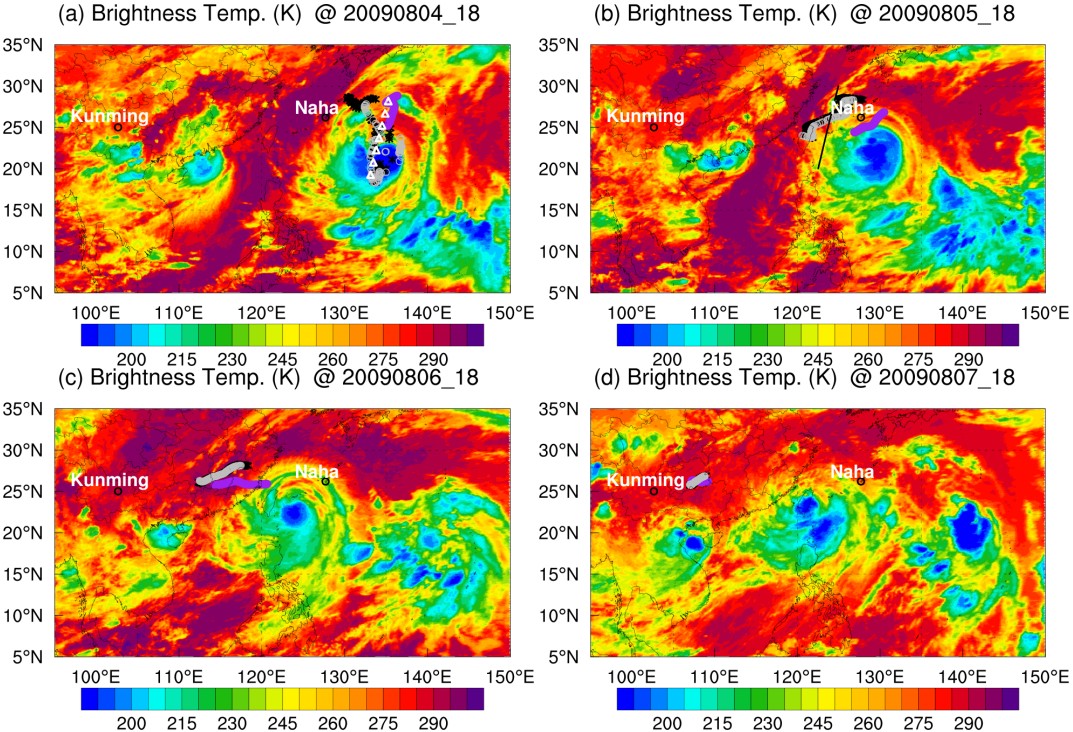

**Figure 7.** Brightness temperature (K) from IR channel of FY-2D on (a) 4 August, 18:00 UTC, (b) 5 August, 18:00 UTC, (c) 6 August, 18:00 UTC, and (d) 7 August 2015, 18:00 UTC with air parcels' locations based on backward trajectories (grey circles–ERA5 diabatic, black asterisk–ERA5 kinematic, purple circles–ERA-Interim diabatic). White triangles mark the MLS tracks (a) and the black line denotes the ground track of CALIPSO (b).

The brightness temperature was interpolated from FY-2D onto the locations of air parcels from the ERA-Interim and ERA5 diabatic trajectory calculations and was used to denote the cloud top temperature. ERA-Interim diabatic trajectories show that air parcels were located right above the deep convection according to the brightness temperature from FY-2D from 5 August, 00:00 UTC to 6 August, 12:00 UTC (Fig. 8a). Compared to ERA-Interim trajectories, ERA5 diabatic trajectory calculations show different results, in particular that the brightness temperature from FY-2D is lower than air parcels' temperature (187 K) on 4 August (Fig. 8b). In other words, the cloud top is higher than air parcels' altitude. The saturation water vapour mixing ratio (SMR) is estimated using the temperature along the backward trajectories of air parcels according to the Goff–Gratch equation (Goff and Gratch, 1946; Murphy and Koop, 2005). The minimum SMR appears on 5 August, at 12:00 UTC, corresponding to the lowest temperature region (Figs. 8c and d). Before air parcels enter the lowest temperature region, several profiles of water vapour from the MLS satellite near to the air parcels' locations (Figs. 7a) are used with mean value of 5±0.3 ppmv marked with a red dot (left-hand side) of Figs. 8c and d on 4 August, 17:30 UTC at 100 hPa. Three days later, the mean water vapour mixing ratio of the same air parcels are about 3 ppmv, marked with red dot (right-hand side). The minimum SMR observed by CFH is approximate 2.7 ppmv in Kunming on 8 August, after air parcels pass through the lowest temperature region from 5 August



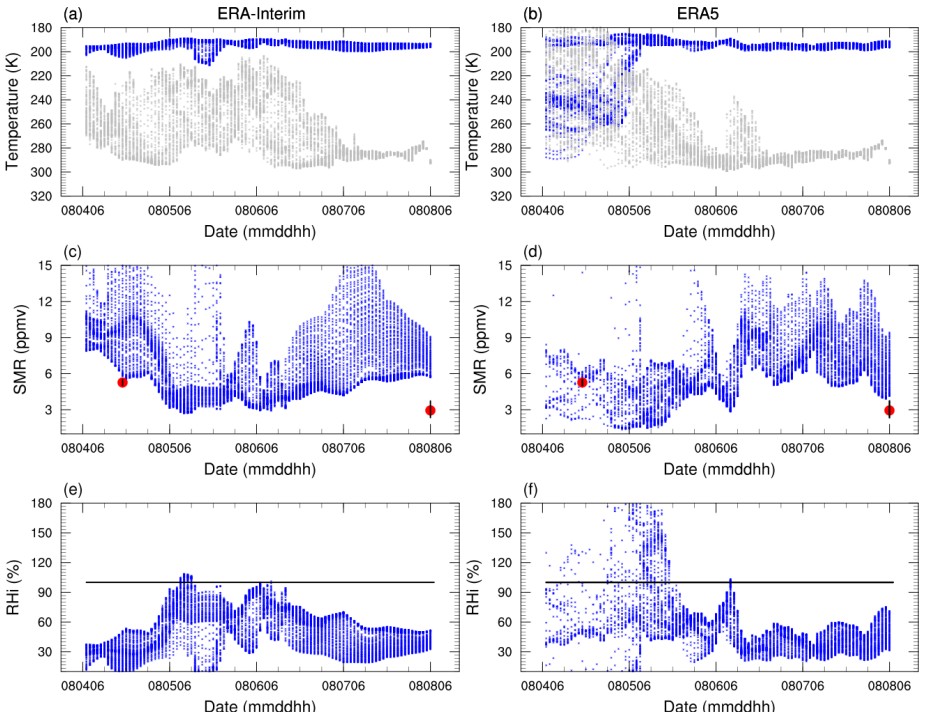

**Figure 8.** The time evolution of (a, b) the temperature (blue dots) and brightness temperature from FY-2D (grey dots), (c, d) the saturation water vapour mixing ratio (SMR), (e, f) relative humidity over ice ($RH_i$) along 4-day backward trajectories of air parcels on 376–384 K from (a, c, and e) ERA-Interim and (b, d, and f) ERA5 diabatic trajectories. The red dots in c and d denote the water vapour mixing ratio from MLS (left-hand side) and balloon observation (right-hand side).

07:00 UTC to 6 August, 12:00 UTC in 2009. Although MLS water vapour profiles usually show mean values of a 3–4 km wide vertical layer, the value corroborates that air parcels become dry, with water vapour mixing ratios decreasing from 5 ppmv to 3 ppmv. The relative humidity over ice ($RH_i$) is calculated by dividing the water vapour mixing ratio by the SMR (Figs. 8e

and f). Air parcels have experienced supersaturation ($RH_i$ up to 180 %) over the lowest temperature region according to ERA5 diabatic trajectories. ERA-Interim diabatic trajectories missed supersaturation, mainly due to higher temperatures along the trajectories of air parcels. Air parcels become dry after they passed through the lowest temperature region. CALIPSO total attenuated backscatter shows cirrus at the altitude of 17 km on 5 August around 18:00 UTC (not shown). The thin cirrus, in turn, provides evidence that dehydration is occurring over the lowest temperature region. As a conclusion, freezing and drying

processes during the lagrangian air parcel history contribute to the low water vapour mixing ratios found over Kunming in August 2009.





## 3.2 Case 2: 10 August 2015

Vertical profiles of temperature, ozone, and water vapour measured over Kunming in August 2015 are shown in Fig. 9. The potential temperature of the lapse rate tropopause on 10 August 2015 is about 383.7 K. The tropopause height in Kunming is
usually higher than in the other regions at the same latitude, based on its location within the ASM anticyclone (Bian et al., 2012). On 10 August, the temperature between 365 K and 376 K shows higher values than on other days. Ozone mixing ratios near the tropopause in August 2015 show a good agreement, except for 10 and 11 August, when much lower ozone mixing ratios are recorded. The ozone vertical profile observed on 10 August 2015 shows extremely low ozone mixing ratios between 364 K and 390 K, with a minimum value of 22 ppbv around 368 K. These low ozone values were also measured on 11 August
2015, with a minimum ozone value of 30 ppbv near 373 K. Low ozone values near the tropopause in 2015 are lower than the values observed on 8 August 2009. The variability of water vapour near the tropopause is smaller, with values from 4 ppmv to 18 ppmv. On 10 August, water vapour mixing ratios are as low as 5 ppmv around 380 K. Unfortunately, water vapour mixing ratios on 11 August are not useful after quality control, because the CFH instrument entered a thick cloud. Therefore, we only focus on the low ozone and low water vapour values (shaded range) near the tropopause on 10 August 2015.

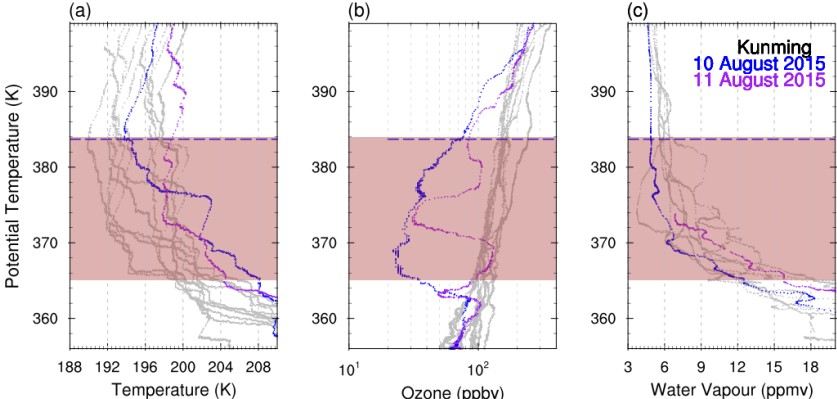

**Figure 9.** As Fig. 2 but for the case in August 2015.

### 3.2.1 Background meteorology

Figure 10 shows the 6-day average (5–10 August 2015) of the geopotential height isolines (>16.7 km) at 100 hPa with the temperature, horizontal wind vector, and the tracks of typhoon Soudelor. Typhoon Soudelor formed as a depression on 29 July 2015 over the middle of the Pacific. It became strong during the period of 3–8 August with reaching peak intensity on 3 August as a Category 5 on the Saffir-Simposon hurricane wind scale. On 9 August 2015, Soudelor degraded to a tropical depression.
Soudelor was the most intense tropical cyclone of the 2015 Pacific typhoon season. The tracks of typhoon Soudelor are just right below the southeast edge of the ASM anticyclone during the period of 6–10 August. This dynamical condition creates



favorable conditions for air parcels on top of the typhoon to be transported into the southern flank of the ASM anticyclonic circulation.

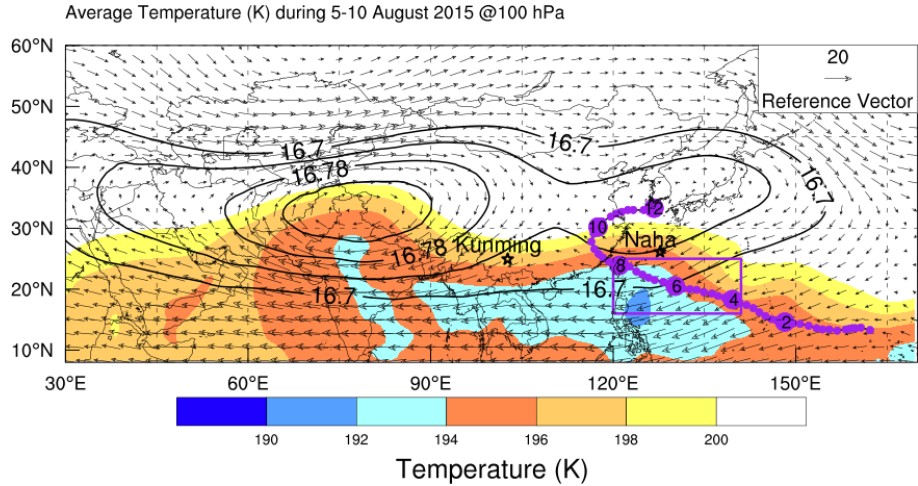

**Figure 10.** As Fig. 3 but for the case in August 2015 and typhoon Soudelor.

### 3.2.2 Low ozone in the tropopause layer

Figure 11a shows mean ozone profile in the rectangle region bounded by purple line in Fig. 10 measured on 31 July, 2, 4, 6, 8, and 10 August from MLS. On 31 July and 2 August, the rectangle region was without any influence of typhoon Soudelor. Ozone profiles from MLS show that the ozone concentration near the tropopause is approximately 50 ppbv on 350 K. In contrast, MLS measurements show that the ozone concentration in the upper troposphere (352–370 K) is about 20 ppbv during the passage of typhoon Soudelor on 4, 6, and 8 August. Ozone values quickly return to a normal value (∼50 ppbv) during the post-typhoon

period on 10 August.

A balloon was launched at Naha site, Japan on 5 August 2015 before typhoon Soudelor passed through. As Fig. 11b shows, low ozone values (22 ppbv) appeared between 352 K and 368 K. This further confirms that deep convection associated with the typhoon Soudelor can uplift air parcels from the boundary layer with low ozone to the tropopause layer. Balloon measurements on 10 August over Kunming captured extremely low ozone between ∼363–382 K (∼14.5–17.5 km) with mixing ratios around

22 ppbv as shown in Figure 11c. Temperatures and ozone from ERA-Interim and ERA5 reanalysis data were interpolated to the balloon ascent profile. The temperatures from ERA-Interim, ERA5, and radiosonde agree very well in the free troposphere and in the lower stratosphere. However, the temperatures from ERA-Interim and ERA5 differ from the radiosonde measurements at the tropopause. The ERA5 ozone profile is in better agreement with ozonesonde measurements in Kunming, especially in the tropopause region, compared to ERA-Interim in August 2015.

A comparison of diabatic and kinematic trajectory calculations is shown based on ERA-Interim and ERA5 reanalysis in Fig. 12. Air parcels from the western Pacific, merged in the easterly wind flow at the southern flank of the ASM anticyclone, via





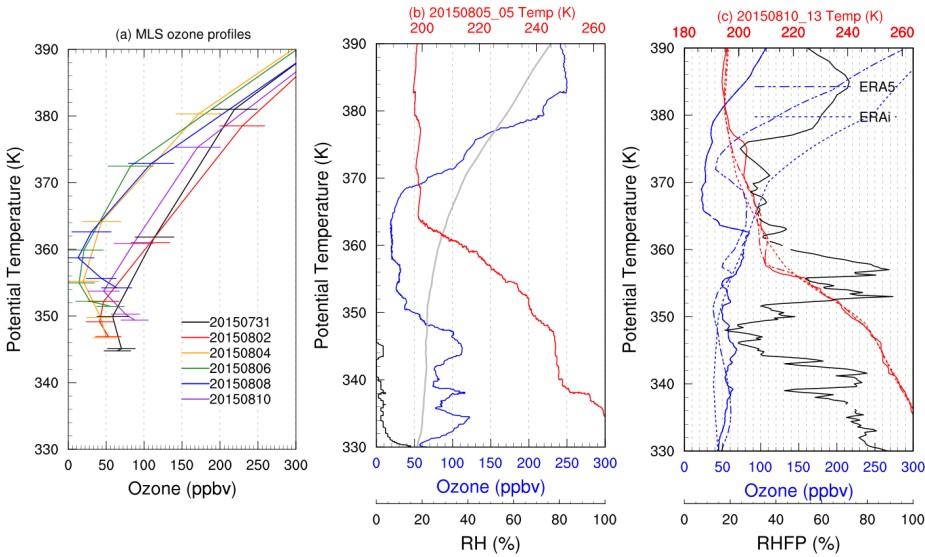

**Figure 11.** (a) Mean ozone profiles measured from MLS satellite data before, during, and after Soudelor pass through the purple rectangle region of Fig. 10. Profiles of ozone (blue), mean ozone (grey), RH (0 black), and temperature (red) in (b) Naha and (c) Kunming. Solid line–observation, dashed line–ERA-interim, dash-dotted line–ERA5.

Naha, Ishigakijima, Laoag, Shantou, Hongkong, Wuzhou, and Haikou, and then were transported to Kunming within 5 days (Figs. 12a–d). Both the kinematic and diabatic trajectories from the ERA-Interim and ERA5 reanalysis data show the lowest temperature region (<190 K) over southern China during the period 7–9 August. Only the ERA5-kinematic trajectory shows

the typical spiral structure of tropical cyclones (Figs. 12c).

The main difference between the backward trajectories originates from the vertical transport over the western Pacific, where typhoon Soudelor occurred (Figs. 12e–h). Although all of the backward trajectory calculations show vertical transport within the deep convection, the timescale and the strength of air parcels' transport are very different. The timescale for the vertical transport from the lower troposphere to the tropopause layer is about 4 days (1–4 August) based on ERA-Interim kinematic

and diabatic trajectories. While the timescale is 2 days (6–7 August) based on the ERA5 kinematic and diabatic trajectories. Both the kinematic and diabatic trajectories from ERA5 show faster vertical transport than ERA-Interim backward trajectories. The time series of the lowest temperature from Haikou, Wuzhou, Hong Kong, Shantou, Laoag, and Ishigakijima are shown based on upper air soundings (Fig. 13). Balloon measurements in Haikou captured the lowest temperature (187 K) during 7–8 August 2015 near the tropopause. Upper air soundings from Ishigakijima, Laoag, Shantou, Hongkong, and Wuzhou also show

the lowest temperature tropopause layer (190–194 K) during 6–10 August 2015.

### 3.2.3   Dehydration based on the ERA-Interim and ERA5 reanalysis data

Figure 14 displays the cloud top temperature (CTT) from FY-2G satellite with air parcels' locations at the corresponding time based on ERA-Interim diabatic trajectory calculations and ERA5 kinematic and diabatic trajectories. On 5 August, Naha was





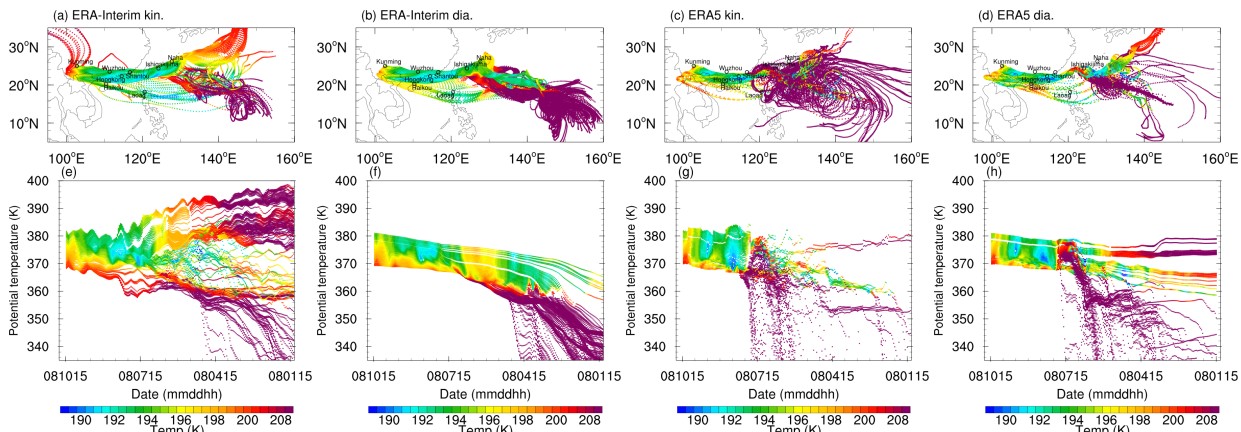

**Figure 12.** As Fig. 5 but for the case on 10 August 2015.

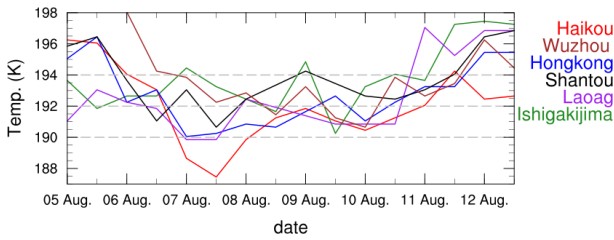

**Figure 13.** As Fig. 6 but for the case in 2015.

located at the northwestern edge of typhoon Soudelor (Fig. 14a). A balloon launched at Naha captured the low ozone mixing
ratios in the TTL as Fig. 11b shows. On 6 August, air parcels from ERA5 kinematic and diabatic trajectory calculations were
located above the center of Soudelor (Fig. 14b). Some of the air parcels from the ERA-Interim diabatic trajectory calculations
still were located at the edge of typhoon Soudelor. Air parcels moved westward toward Kunming on 8 August (Fig. 14c),
and CTT show the lowest temperature (190 K) above Taiwan. On 9 August, air parcels arrived in Kunming under clear sky
conditions (Fig. 14d).
Figures 15a and b show the temperature along the 4-day backward trajectories for air parcels at 370–381 K on 10 August
2015 with CTT from FY-2G. From 00:00 UTC on 7 August to 12:00 UTC on 8 August, the air parcels are located at the top
of the deep convection associated with typhoon Soudelor according to the CTT of FY-2G (Figs. 15a and b). Before air parcels
enter the lowest temperature region, the water vapour mixing ratios from the MLS satellite are $7\pm0.3$ ppmv at 05:30 UTC on
8 August at 100 hPa (Figs. 15c and d). Two days later, the water vapour mixing ratios observed by CFH are approximately
5 ppmv in Kunming (red dot on right-hand side), hence have significantly decreased after air parcels passed through the lowest
temperature region from 05:00 UTC to 21:00 UTC on 8 August 2015. The $RH_i$ is calculated according to water vapour mixing
ratios (CFH observed on 10 August) divided by the SMR (Figs. 15e and f). Air parcels experienced supersaturation during





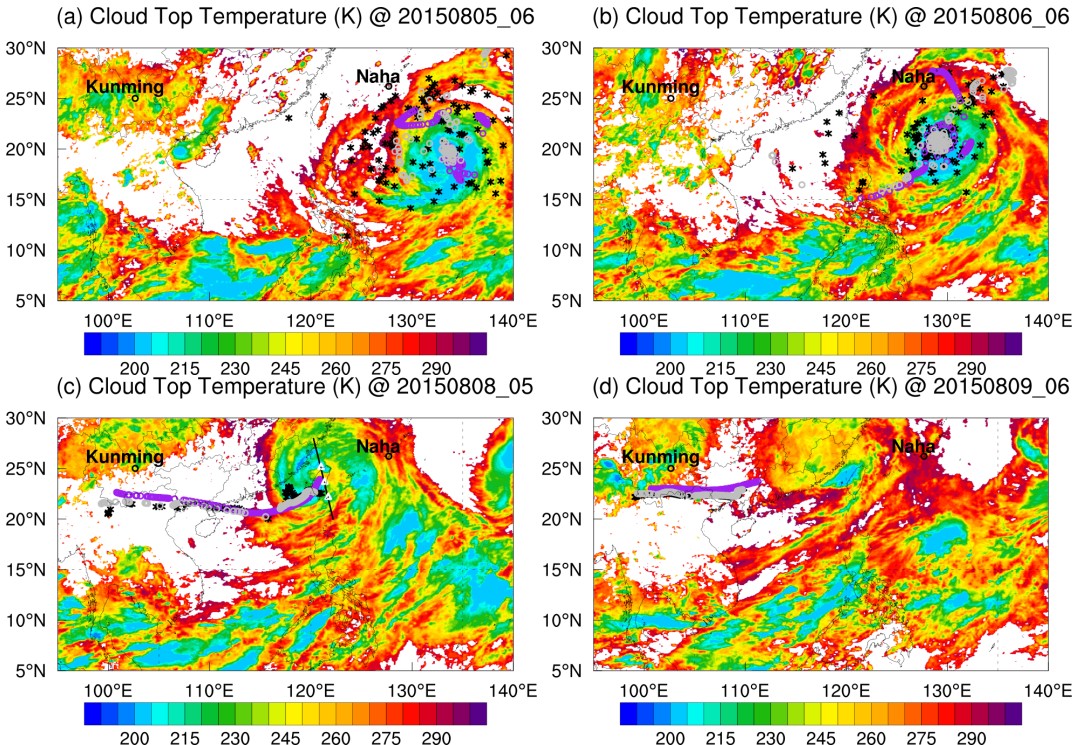

**Figure 14.** Cloud top temperature from FY-2G satellite on (a) 5 August, (b) 6 August, (c) 8 August, and (d) 9 August 2015 with backward trajectories (grey circles–ERA5 diabatic, black asterisk–ERA5 diabatic, purple circles–ERA-Interim diabatic) for air parcels with low ozone and low water vapour.

the lowest temperature period based on ERA5 diabatic trajectory calculations. Freezing and drying processes contribute to the variability of water vapour mixing ratios observed over Kunming in August 2015 again.

## 4  Discussion and conclusions

The low ozone and low water vapour mixing ratios near the tropopause measured on 8 August 2009 and 10 August 2015 in Kunming are investigated using balloon measurements, satellite measurements, and CLaMS simulations. MLS ozone and water vapour measurements and trajectory calculations from the CLaMS model confirm that the vertical transport is largely caused by tropical cyclones and the horizontal transport is caused by the ASM anticyclone. The interplay between tropical cyclones and the ASM anticyclone exerts a major influence on transporting the ozone-poor western Pacific boundary air to the tropopause layer and even to the ASM anticyclone region. This interplay is consistent with a former study by Li et al. (2017) analysing low ozone values in Lhasa measured in August 2013. The deep convective clouds associated with tropical cyclones have considerable implications on ozone in the UTLS (Fu et al., 2013; Li et al., 2017) and in consequence other chemical species, such as hydroxyl radical (Rex et al., 2014).





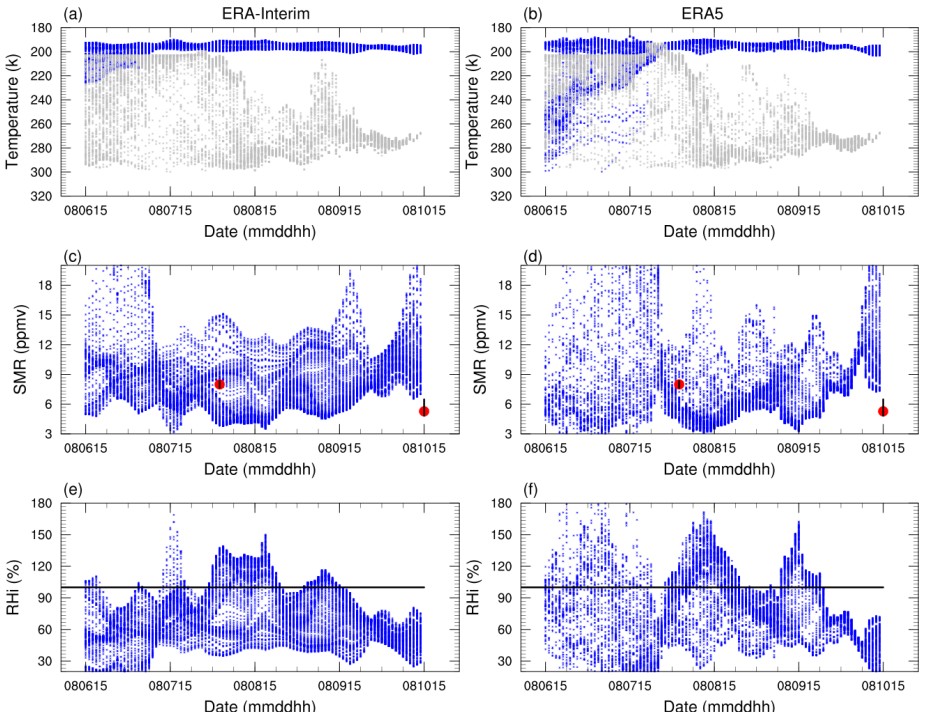

**Figure 15.** As Fig. 8 but for the case on 10 August 2015.

Besides tropical cyclones, other potential mechanisms (wave activity) can also contribute to dehydration in the tropopause
layer (Fujiwara et al., 2012). The lowest temperatures at the tropopause over the western Pacific ocean mainly drive the dehy-
dration process. As a result, air parcels become dry when they pass the low temperature region before entering the stratosphere.
Holton and Gettelman (2001) call that "the cold trap in winter". Our case study for the summer season is consistent with this
low temperature picture, with lowest temperatures also occurring over the western Pacific and at the southeastern edge of the
ASM, however, further north compared to winter condition.

Our observations confirm previous studies (Hasebe et al., 2013; Inai et al., 2013; Wright et al., 2011), that air masses
emanating from the South China and Philippine Sea become particularly dry when they pass through the lowest temperature
regions around the tropopause over the western Pacific. The easterly winds on the southern flank of the ASM anticyclone
transport these air masses with low ozone and low water vapour to the west over a distance of approximately 2,000 km to
Kunming.

The interplay between tropical cyclones and the ASM anticyclone has the potential to impact the long term trends of ozone,
water vapour, and even the optically thin cirrus near the tropopause, particularly under climate change conditions, when the
occurrence of tropical cyclones will be more frequently.





The trajectory calculations using ERA5 data show faster and stronger vertical transport than ERA-Interim reanalysis data.
The ERA5 wind field appears to represent convective updrafts and tropical cyclones well, due to ERA5's better spatial and temporal resolution. This is consistent with the stronger vertical transport in ERA5 compared to ERA-Interim reanalysis data.

*Code and data availability.* ERA-Interim and ERA5 meteorological reanalysis data are free available from the web page: http://apps.ecmwf. int/datasets/ (last access: 3 August 2018). The MLS ozone and water vapour data were download from https://acdisc.gesdisc.eosdis.nasa.gov/ data/ (last access: 3 November 2017). The FY-2D and FY-2G data used in this study can be obtained at http://satellite.nsmc.org.cn/PortalSite/
Data/Satellite.aspx (last access: 14 June 2018). The ozone profiles for Naha are provided on the World Ozone and Ultraviolet Radiation Data Centre (WOUDC) https://woudc.org/data/explore.php?lang=en (last access: 15 June 2018). The upper air soundings data were download from the University of Wyoming http://weather.uwyo.edu/upperair/sounding.html (last access: 12 September 2018). The SWOP data of this paper are available upon request to Jianchun Bian (bjc@mail.iap.ac.cn). The CLaMS backward trajectories calculations may be requested from Dan Li (lidan@mail.iap.ac.cn). The CLaMS model code can be requested from Dr. Rolf Müller (ro.mueller@fz-juelich.de)

**Appendix A: Comparisons of ERA-Interim and ERA5 wind fields**

Figure A1 compares the geopotential height and horizontal wind speeds at 150 hPa on 4, 5, and 6 August 2009 between ERA-Interim and ERA5 reanalysis data. The geopotential height from ERA-Interim and ERA5 data shows a similar pattern. The horizontal wind fields also show the same pattern, but the ERA5 wind fields display more small-scale structures.



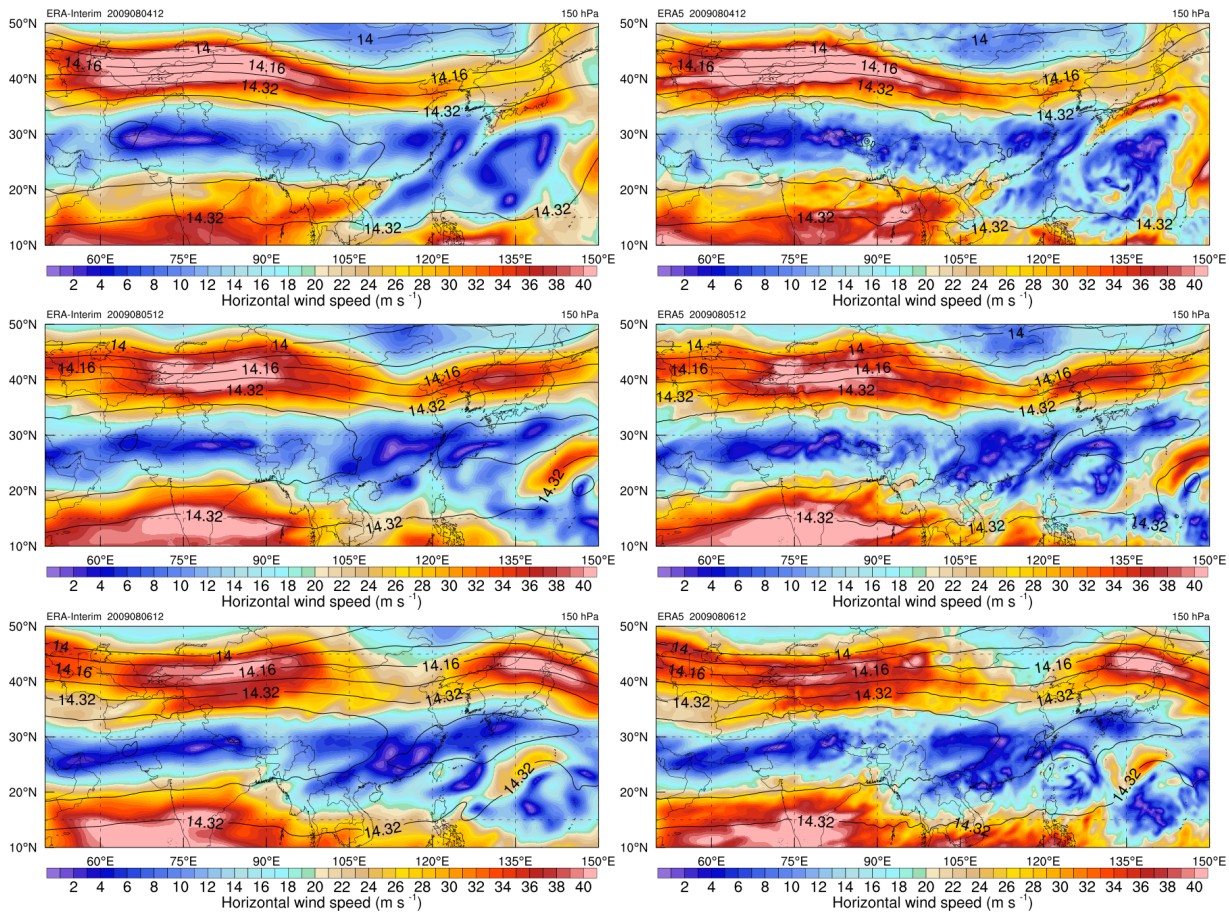

**Figure A1.** Geopotential height (black lines, gpkm) and horizontal wind speeds (shaded, $\mathrm{m\,s^{-1}}$) at 150 hPa on 4, 5, and 6 August 2009 from ERA-Interim (left) and ERA5 (right).





*Author contributions.*  DL prepared for the first draft. BV, RM, GG, FP, and MR provided effective and constructive comments and helped improve the paper. JB and HV gave useful comments. JB, ZB, QL, JZ, and DL made the balloon-borne measurements in Kunming. HV supported for the measurements.

*Competing interests.*  The authors declare that they have no conflict of interest

*Acknowledgements.*  This research was supported by the National Key Research and Development program of China (2018YFC1505703) and the National Science Foundation of China (Grant No. 91837311, 41975050, 41605025, 41675040, 91637104, and 41705127). Our activities contribute to the European Community's Seventh Framework Programme (FP7/2007-2013) as part of the StratoClim project (Grant agreement No. 603557). The authors thank the Strategic Priority Research Program of Chinese Academy of Sciences (Grant No. XDA17010102), the International Postdoctoral Exchange Fellowship Program 2017 under grant No. 20171015 and China Postdoctoral Science Foundation (2015M581153).





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
