# Peer review of "Dehydration and low ozone in the tropopause layer over the Asian monsoon caused by tropical cyclones: Lagrangian transport calculations using ERA-Interim and ERA5 reanalysis data"

_Atmospheric Chemistry and Physics, 2019_

## Referee Comment (RC1) · Anonymous Referee #1 · 22 Oct 2019

This paper show low ozone and water vapor near the tropopause at Kunming, China, from Balloonsonde measurements. Multiple data sets support these results, e.g., Feng Yun-2D, FengYun-2G, Aura Microwave Limb Sounder (MLS) satellite data. The observed low ozone and low water vapor are attributed to transport of boundary layer air parcels originated from deep convection region associated with tropical cyclones in the western Pacific. Then dehydration of air parcels happens while passing through the coldest temperature region in the UTLS. Dehydration is also linked to convective clouds associated with the cyclone in the pacific. The results also highlight that vertical

transport is faster and stronger in ERA5 than ERA-Interim. This situation was observed in two cases 1-8 August 2009 and 10th August 2015.

The Authors have presented the results with sufficient pieces of evidence and justifications. The manuscript should be published in the ACP after the revision.

I have the following few suggestions which authors may consider to include during revision.

In the years 2009 and 2015, there was El Niño in the pacific. Don't you think that vertical transport of air parcels will be influenced by warming at the onset phase of El Niño?

Linkages of dehydration of air mass with convective clouds in the Pacific associated with the cyclone and passage through coldest temperature region are not clear. It should be stated clearly in the discussion section and elsewhere.

It is excepted that the marine boundary layer containing low ozone and high water vapor may get dehydrated near the cold tropopause. Authors have cited several studies reporting poor ozone air arriving near the tropopause from the tropical marine boundary layer, and there is cold trap dehydration near the TTL (page 2 Lines2-50). What are the new results?

Authors should highlight the implications of low ozone and low water vapor in the abstract and conclusions.

Do you see low ozone and low water vapor values during the passage of cyclone, and low values disappears after that? Daily plots from 7- 13 August 2009 and 3-18 August 2015 will be helpful.

Dehydration and ice formation can be shown in a figure from ERA5 and ERA-Interim to support the results.

The authors should show a vertical profile-plot showing the difference between ozone

on 8 August 2009 and daily climatological mean. A similar plot for water vapor will be useful. Also, similar plots for ozone and water vapor on 10 August 2015 should be shown in supplementary figures. It will help to quantify the decrease in ozone and water vapor at different altitudes on the respective days.

Do you have observations that show low ozone and water vapor outside the cyclone days? Is it a regular feature near the TTL over China during August? Or is there stronger dehydration during cyclone days due to clouds (cumulonimbus) reaching the tropopause?

Page 11 L 212-215: These statements are confusing. What is the reason for the dehydration of air parcel? Is it related to the deep convective clouds or due to the passage of air parcels through cirrus clouds?

One can use brightness temperature and optical thickness, both, to determine the presence of convective cloud or cirrus cloud.

The method adopted for identification of convective cloud should be mentioned in section 3.1.1.

Figure-1 shows geopotential height at 150 on 4,5, and 6 August 2009. Why not during all the days from 4-8 August 2009?

Page 5-6: The outflow of deep convection is stronger on 6 August 2009 while the vertical profile of ozone and water vapor shows a low amount on 8 August 2009, Why?

Profiles of ozone show minimum ozone at a different height on 4th and 8 August 2009. Is it due to the distance of a cyclone from the observational site?

Page 12 Case 2: 10 August.

On 3 August 2015, intensity typhoon Soudelor reached Category 5. The authors should state the reason for showing analysis on 10 August 2015 when the typhoon was degraded to depression. Balloon measurements are available from 3 to 18 August 2015

(Page 3, L74). The authors should present a case for any day during 2-3 August 2015.

Page 7, L 147, and Page 13L 246: If authors want to show vertical profiles which has no influence of cyclone, then profiles should be presented for the days before the formation of cyclone in the Pacific. Profiles on 4 August 2009 and 5 August 2015 are influenced by the cyclonic winds since on these days Morakot, and Soudelor cyclones were passing through the Pacific. If authors want to show the influence of cyclones, then profiles should be shown on the same days at Naha and Kunming. Is there any reason for choosing different days? If so, please clarify.

ASM anticyclone is associated with a large amount of water vapor and low ozone. However, authors want to show that deep convective clouds associated with cyclone, which reaches near the tropopause (cold cloud tops), cause freeze-drying at the outflow of ice clouds. It should be made clear in the abstract and conclusions.

Figures: fonts of X-axis and Y-axis in all the figures should be bigger and bold.

---

## Referee Comment (RC2) · Anonymous Referee #2 · 24 Oct 2019

General comments:

This study presents low ozone and low water vapor layers observed in the tropopause layer over Kunming, southern China and discusses them from the perspective of air mass transport and dehydration. The balloon-borne observation results are unique and interesting. The analysis method combining satellite measurement data with trajectories driven by state-of-the-art reanalysis data is adequate and persuasive. The results are impressively presented and it seems to be fascinating for wide range of readers, even though the study is based only on two case studies. In my view, the

paper should be published in ACP after minor revision. Some specific comments and suggestions are listed below, they are all minor.

Specific comments:

- Line 104–106: What is definitional or calculational difference between TBB and CTT? How about making a brief explanation of the two, here. Further, are the coloring somewhat different between Figs. 7 and 14? Region where > 295 K is colored in deep red for Fig. 7, but in white for Fig. 14?

- Section 3.1: How about adding the definition or citation for "lapse rate tropopause."

- Line 204–206: The authors describe "Three days later, the mean water vapour mixing ratio of the same air parcels are about 3 ppmv, marked with red dot (right-hand side). The minimum SMR observed by CFH is approximate 2.7 ppmv in Kunming on 8 August," From this statement, I expect that the supersaturation had been observed because the water vapor mixing ratio is higher than the SMR. However, the RHFP shown in Figure 4 is lower than 100% through the whole altitude. Is the "SMR" in line 205 a mistake of "water vapor mixing ratio"?

- Line 209–211: I feel the descriptions are a little sudden. How about writing a little more particularly about what you are assuming and what you are comparing? For example, how about adding a statement, such as "If we assume that the air mass retains water vapor mixing ratio when it had been observed by the CFH," at the beginning of the sentence "air parcels have experienced supersaturation (RHi up to 180%) over ..."

In addition, the following is just a suggestion, but if the Lagrangian minimum SMR (SMRmin_i) that the each trajectory (trj_i) has experienced after the final convective encountering (latest Tbb < T_trj) is estimated, and the average SMRmin of all trajectories is calculated, can it make additional discussion or provide interesting insight by comparing the average SMRmin with the water vapor mixing ratio observed by CFH? Also the case 2 is.

---

## Referee Comment (RC3) · Anonymous Referee #3 · 11 Nov 2019

The manuscript addresses the problem of the lifting of air by a tropical cyclone in the West Pacific during the Asina Monson and the resulting dehydration and low ozone layer in the lower stratosphere. The study consists in two case studies. It uses data from ozone and water vapor balloon profiles and MS data. Another interest is in the comparison of the new ERA5 and the ERA-Interim regarding the Lagrangian trajectories near to a cyclone.

This work is interesting but somewhat misses to provide necessary details and I find the authors could have gone deeper into the analysis.

In 2.3 Provide information about the vertical sampling of the vertical interval and mention that the total diabatic heating rates, including latent heating are used (as it seems the case from the results).

First case

There is a layer of low ozone and low water vapor on 8 August near 365 K which could be a remain of that of the 4 August but no attention has been paid to it. The paper would be strengthened by showing the origin of that layer.

Lines 197-198: It seems that ERA5 predicts that the trajectories where inside the clouds but this fact is not exploited and the parcels are treated as unsaturated in the sequel.

What means, on line 203 that MLS water vapor is retrieved near the parcels location? The MLS value is the same for both ERA5 and ERA-Interim while the parcels are not located at the same level and can MLS see through the cloud anvils under which are located the ERA5 parcels? Therefore, how this value can be used as a reference to evaluate the dehydration? This is a crucial point in the analysis that deserves better description and justification.

Line 209: It is unclear which water vapor is used for this estimate of the relative humidity. Is it the from the balloon flight of 8 August?

On line 212, the authors mention that parcels get dry after passing through the lowest temperature region, but not attempt is made to see whether this can explain quantitatively the observed dehydration, even using a simple freeze-drying process relaxing to saturation. The analysis is too qualitative on this point (provided the previous question is also correctly answered).

I find a bit confusing that the dates are oriented in opposite directions in figs. 4 and 8 on one side and 12 and 15 on the other side.

Second case

Here the MLS water vapor reference is retrieved at a date posterior to the exit of ERA5 trajectories from the clouds but details are still missing and again no attempt are made to quantitatively explain the dehydration.

Looking at trajectories in fig.12, I do not find a convincing rise of 15 K of the low ozone layer between 5 and 10 August as required by fig. 11(b,c).

Why trajectories are stopped at 330 K instead of 350 K as in the first case is not explained.

---

## Author Response (AR1)

**Authors Reply to Anonymous Referee #1**

45

This paper show low ozone and water vapor near the tropopause at Kunming, China, from Balloonsonde measurements. Multiple data sets support these results, e.g., Feng Yun-2D, FengYun-2G, Aura Microwave Limb Sounder (MLS) satellite data.

- 5 The observed low ozone and low water vapor are attributed to transport of boundary layer air parcels originated from deep convection region associated with tropical cyclones in the western Pacific. Then dehydration of air parcels happens while passing through the coldest temperature region in the UTLS. Dehydration is also linked to convective clouds associated with the cyclone in the pacific. The results also highlight that vertical transport is faster and stronger in ERA5 than ERA-Interim. This situation was observed in two cases 1-8 August 2009 and 10th August 2015.
- 10 *The Authors have presented the results with sufficient pieces of evidence and justifications. The manuscript should be published in the ACP after the revision.*

I have the following few suggestions which authors may consider to include during revision.

**Reply:** The authors appreciate the anonymous referee #1 for your constructive and meaningful comments which are very helpful for our manuscript. Please see below our responses point by point (*referees' comments in blue with italics font* and our response in black). Further, the revised manuscript is added highlighting in colour any changes compared to the ACPD version.

**In the years 2009 and 2015, there was El Niño in the pacific. Don't you think that vertical transport of air parcels will be influenced by warming at the onset phase of El Niño?**

- 20 Reply: The long term ozone trend in the tropical stratosphere (17–21 km) is primarily linked to the QBO and ENSO (Randel and Thompson, 2011). ENSO variability in ozone is about 10% relative to local background levels. The ozone change is linked to enhanced zonal mean tropical stratospheric upwelling in ENSO warm events (Randel et al., 2009). We check the oceanic El Niño index from the webpage https://ggweather.com/enso/oni.htm . The 3-month (JJA) mean index is 0.5 in 2009 and 1.8 in 2015, respectively. There is a very strong El Niño in 2015. The ozone negative variability on 8 August 2009 and 2015 are
- 25 about 60%. Ozone profiles without influence of typhoon in August of 2009 and 2015 show small variability (Figs. 2b and 9b in the revised manuscript). In 2013, the 3-month mean Oceanic Niño index value is -0.4. Li et al. (2017) show three tropical cyclones have uplifted the marine boundary air with low ozone to the tropical tropopause layer in August 2013. The ozone vertical transport within several days is strongly associated with the intense typhoon case. El Niño has in general the potential to impact vertical upward transport in the Pacific, however, the contribution to strong uplift within a single typhoon is to be
- 30 assumed to be small. Further it is hard to differentiate between the impact of El Nino and tropical cyclones on vertical upward transport of air masses.

**Linkages of dehydration of air mass with convective clouds in the Pacific associated with the cyclone and passage through coldest temperature region are not clear. It should be stated clearly in the discussion section and elsewhere.**

- 35 **Reply:** We rewrite the sentence in the conclusions from "...Our case study for the summer season is consistent with this low temperature picture, with lowest temperatures also occurring over the western Pacific and at the southeastern edge of the ASM, however, further north compared to winter condition." to "...Our case studies show that the low temperature (

Figure 1. The variability of ozone and water vapour in August 2009 (top) and 2015 (bottom) deduced from balloon measurements in Kunming.

channel infrared (11 $\mu$ m) window area retrieval (?). Cloud radiating temperature is assumed to be  $B(T) = (1 - \varepsilon)B(T_{cs}) + \varepsilon B(T_c)$ .

Where B is the Planck function.  $\varepsilon$  is the effective emissivity (?). CTT is computed with the cloud-top emissivity parameter  $(\varepsilon_t)$ . For thick clouds,  $\varepsilon_t = \varepsilon$ .

For thin clouds,  $\varepsilon_t$  depends on  $T_c$ ,

if  $T_c < 245$  K then  $\varepsilon_t = (-0.00914T_c + 2.966)\varepsilon$ ,

105 if  $T_c > 280$  K then  $\varepsilon_t = (0.00753T_c + 1.12)\varepsilon$ .

The cloud-top temperature is

 $T_t = B^{-1}\{[B(T) - (1 - \varepsilon_t)B(T_{cs})] \lor \varepsilon_t\}.$

TBB>295 K in Fig. 7 is blackbody temperature, CTT>295 K in white for Fig. 14 means clear sky. To be consistent, we used FY-2G TBB instead of the CTT in the revised manuscript.".

110

115

*Figure-1 shows geopotential height at 150 hPa on 4, 5, and 6 August 2009. Why not during all the days from 4–8 August 2009?* **Reply:** We added the geopotential height at 150 hPa on 7 and 8 August 2009 to Fig. 1.

Page 5–6: The outflow of deep convection is stronger on 6 August 2009 while the vertical profile of ozone and water vapor shows a low amount on 8 August 2009, Why?

**Reply:** Sure, the convective outflow is strongest on 6 August around Naha over the western Pacific. The transport of these air masses from Naha to Kunming over the Tibetan Plateau needs 2 days, therefore the profiles of water vapour and ozone in Kunming have low values on 8 August 2009.